# Gomisin N from *Schisandra chinensis* Ameliorates Lipid Accumulation and Induces a Brown Fat-Like Phenotype through AMP-Activated Protein Kinase in 3T3-L1 Adipocytes

**DOI:** 10.3390/ijms21062153

**Published:** 2020-03-20

**Authors:** Kippeum Lee, Yeon-Joo Lee, Kui-Jin Kim, Sungwoo Chei, Heegu Jin, Hyun-Ji Oh, Boo-Yong Lee

**Affiliations:** Department of Food Science and Biotechnology, College of Life Science, CHA University, Seongnam, Kyonggi-do 13488, Korea; joy4917@hanmail.net (K.L.); joy4917@naver.com (Y.-J.L.); kuijin.kim@gmail.com (K.-J.K.); sungwoochei@gmail.com (S.C.); heegu94@hanmail.net (H.J.); guswl264@naver.com (H.-J.O.)

**Keywords:** gomisin N, 3T3-L1, lipid accumulation, browning, obesity

## Abstract

Obesity results from an imbalance between energy intake and energy expenditure, in which excess fat is stored as triglycerides (TGs) in white adipocytes. Recent studies have explored the anti-obesity effects of certain edible phytochemicals, which suppress TG accumulation and stimulate a brown adipocyte-like phenotype in white adipocytes. Gomisin N (GN) is an important bioactive component of *Schisandra chinensis*, a woody plant endemic to Asia. GN has antioxidant, anti-inflammatory and hepatoprotective effects in vivo and in vitro. However, the anti-obesity effects of GN in lipid metabolism and adipocyte browning have not yet been investigated. In the present study, we aimed to determine whether GN suppresses lipid accumulation and regulates energy metabolism, potentially via AMP-activated protein kinase (AMPK), in 3T3-L1 adipocytes. Our findings demonstrate that GN inhibited adipogenesis and lipogenesis in adipocyte differentiation. Also, GN not only increased the expression of thermogenic factors, including uncoupling protein 1 (UCP1), but also enhanced fatty acid oxidation (FAO) in 3T3-L1 cells. Therefore, GN may have a therapeutic benefit as a promising natural agent to combat obesity.

## 1. Introduction

Obesity, a widespread major health problem, is associated with metabolic disease [1]. Especially, numerous studies have established that abnormal visceral fat accumulation is accompanied by elevated metabolic risk factors including diabetes, dyslipidemia and cardiovascular disease [2]. Obesity is characterized by growing adipose tissue masses with increased numbers (hyperplasia) and size (hypertrophy) of adipose cells. Adipose tissues play a central role in regulating metabolic homeostasis [3]. In mammals, adipose tissues are divided into two major types: white adipose tissue (WAT) and brown adipose tissue (BAT) [4]. WAT stores excessive energy in the form of triglycerides (TGs) [5]. In contrast, BAT is a thermoregulatory organ. This metabolically active organ converts energy into heat via fatty acid oxidation (FAO) [6]. Recent studies revealed that brown adipocytes are also present in WAT, known as brown-like adipocytes or beige adipocytes [7]. Adipose browning stimuli alter expression of proteins involved in brown-like adipocyte formation, including PR domain containing 16 (PRDM16), peroxisome proliferator-activated receptor gamma co-activator 1 alpha (PGC1α) and uncoupling protein 1 (UCP1) [7,8]. Brown and brown-like adipocytes have similar characteristics, including multilocular lipid droplet structure, high mitochondrial content and high expression of brown adipocyte-specific factors, especially UCP1. Our previous studies have identified that nutritional agents can induce a brown-like phenotype in white adipocytes, which elevates mitochondrial biogenesis and thermogenic capacity [9]. Stimulation of WAT browning is a potential therapeutic approach for obesity and metabolic disease.

Lipid metabolism has been targeted for the treatment of obesity and obesity-related metabolic disease [10]. Adipogenesis is a complex mechanism of adipocyte differentiation from pre-adipocytes, and involves various transcription factors, such as peroxisome proliferator-activated receptor gamma (PPARγ), CCAAT enhancer-binding protein alpha (C/EBPα) and fatty acid synthase (FAS) [11]. Adipocytes store excess energy as TGs, which are produced by lipogenic enzymes. This process is regulated by TG synthesis factors, including lysophosphatidic acid acyltransferase theta (LPAATθ), diacylglycerol acyltransferase 1 (DGAT1) and phosphatidate phosphatase (LPIN1) [12].

AMP-activated protein kinase (AMPK) is a central regulator of energy metabolism and mitochondrial biogenesis in adipocytes [13]. AMPK activation deactivates acetyl-CoA carboxylase (ACC), resulting in decreased fatty acid synthesis and increased FAO [14]. ACC promotes intracellular lipid synthesis in late stage adipogenesis. Thus, regulation of adipogenic transcription factors is important in attenuating adipocyte differentiation in obesity. Also, AMPK activation elevates mitochondrial oxidative capacity and expression of factors involved in thermogenesis, such as PRDM16, PGC1α and UCP1 [15,16]. Recent studies suggest AMPK is necessary for BAT development in obese mice, which is associated with expression of PRDM16. In addition, PGC1α is induced not only by PRDM16, but also by AMPK activation [17,18].

*Schisandra chinensis* (SC) is widely used as a traditional herbal sedative and tonic agent in China, Korea and Japan. Gomisin N (GN) is a physiological lignin isolated from SC, which contains bioactive components such as lignans, triterpenoids and polysaccharides [19]. Recent studies reported that lignans isolated from SC possess anti-oxidant, anti-inflammatory, anti-cancer and hepatoprotective properties [20,21,22,23]. Other studies determined that these herbal agents confer cardioprotection against myocardial ischemia-reperfusion injury by attenuating oxidative myocardial damage in rats [24], and also possess anti-hypertensive, neuroprotective and hepatoprotective effects in both in vitro and in vivo models [25,26,27]. Lastly, GN is reported to inhibit lipid accumulation in 3T3-L1 cells by suppressing cell proliferation and cell cycle progression in early stages of adipogenesis, preventing obesity and ameliorating hepatic steatosis in high-fat diet-induced obese mice [28]. However, the effects of GN on thermogenic activity and FAO and subsequent inhibition of lipid accumulation in adipocytes have not been explored. Therefore, the present study sought to elucidate the effect of GN on lipid accumulation and adipocyte browning during adipocyte differentiation. 

## 2. Results

### 2.1. GN Decreases Lipid Accumulation by Inhibiting Adipogenesis

To investigate the anti-obesity effects of GN in adipocytes, 3T3-L1 cells were differentiated with 0, 6.25, 12.5 and 25 μM GN for 8 days. Oil red O (ORO) staining revealed that GN significantly decreased fat accumulation in 3T3-L1 cells (Figure 1A,B). We performed Western blot analysis to determine the mechanisms for GN suppression of lipid accumulation. We determined whether GN regulated expression of adipogenic factors such as PPARγ, C/EBPα and FAS. Control-differentiated (CD) cells exhibited higher expression levels of adipogenic proteins relative to non-differentiated (ND) cells, including PPARγ, C/EBPα and FAS, which was reversed by GN in a dose-dependent manner (Figure 1C,D). Treatment with 25 μM GN markedly reduced PPARγ by 42.3%, C/EBPα by 20.8% and FAS by 33.4%, compared with CD cells. In addition, mRNA levels of adipogenic genes including *Pparγ*, *C/Ebpα* and fatty acid-binding protein 4 (*Fabp4*) were significantly decreased by 25 μM GN treatment in 3T3-L1 cells. Lastly, cell viability of 3T3-L1 cells treated with GN was evaluated using an XTT assay. GN was non-cytotoxic at lower concentrations but was cytotoxic at 50 μM (Figure 1F). Based on these data, subsequent experiments used GN concentrations up to 25 μM. Taken together, these findings demonstrated that GN suppressed lipid accumulation in 3T3-L1 adipocytes by attenuating expression of adipogenic factors.

### 2.2. GN Inhibits TG Synthases by Attenuating Expression of Lipogenic Factors

To determine the effect of GN on TG synthesis, we measured intracellular TG accumulation in 3T3-L1 cells using a commercial TG assay kit. GN effectively reduced the TG contents of 3T3-L1 cells (Figure 2A). A recent study reported that decreasing PPARγ expression suppresses expression of sterol regulatory element-binding transcription factor 1c (SREBP1c), which plays a key role in TG biosynthesis [29]. In the present study, GN dose-dependently reduced SREBP1c expression in 3T3-L1 cells (Figure 2B,C). In addition, GN significantly decreased the lipogenic proteins LPAATθ, DGAT1 and LPIN1 (Figure 2B,C). In other hands, GN effectively increased the phosphorylated hormone sensitive lipase (p-HSL) expression, which catalyzes the hydrolysis of cellular triacylglycerol and diacylglycerol in adipocytes. These finding indicated the GN could promote hydrolysis of TGs to glycerol and free fatty acids by downregulating lipogenic factors. 

### 2.3. GN Enhances Thermogenic and FAO Markers in 3T3-L1 Cells

To evaluate the browning effect of GN, 3T3-L1 cells were treated with various concentrations of GN (6.25, 12.5 or 25 μM) during differentiation, and protein phosphorylation/levels of thermogenic markers were then measured by Western blotting. GN treatment significantly increased phosphorylated AMPK (p-AMPK) and BAT markers, including fibroblast growth factor (FGF21), PRDM16, PGC1α and UCP1 (Figure 3A,B). Moreover, treatment with 25 μM GN up-regulated mRNA levels of FAO genes, such as peroxisome proliferator-activated receptor alpha (*Pparα*), peroxisome proliferator-activated receptor delta (*Pparδ*), carnitine palmitoyltransferase 1 (*Cpt1*) and carnitine palmitoyltransferase 2 (*Cpt2*). A recent study suggested that UCP1 in mitochondria inner membranes plays an important role in energy consumption [30,31]. We analyzed differentiated cells treated with GN using immunostaining. MitoTracker Red staining, an indicator of mitochondrial activity and UCP1 expression (conjugated with FITC green) were increased by 25 μM GN in a dose-dependent manner. These changes to a BAT-like phenotype are generally associated with increased metabolic rate (Figure 3D).

### 2.4. Role of AMPK in GN Modulation of ACC and UCP Expression

AMPK is a master regulator of energy homeostasis and mitochondrial FAO in adipose tissue [32]. In this study, to investigate the role of AMPK in GN modulation of lipid metabolism, we used the AMPK activator 5-aminoimidazole-4-carboxamide ribonucleotide (AICAR, 10 μM) and AMPK inhibitor 6-[4-(2-Piperidin-1-yl-ethoxy)-phenyl)]-3-pyridin-4-yl-pyrrazolo [1,5-a]-pyrimidine (Compound C, CC, 10 μM). AICAR treatment increased phosphorylation of AMPK and its downstream target ACC in adipocytes, as did 25 μM GN (Figure 4A,B). ACC catalyzes production of malonyl-CoA, inhibiting lipogenesis and CTP1 expression. The phosphorylation of ACC by AMPK activation results in inactivation of ACC, so therefore CPT1 expression can be recovered and FAO promoted [33]. On the other hand, CC decreased p-AMPK and p-ACC, which was partially recovered by GN co-treatment (Figure 4C,D). 

### 2.5. GN Regulates Lipid Metabolism and Induces 3T3-L1 Browning

To compare the browning effect of GN with that of the well-known browning inducer bone morphogenetic Protein 4 (BMP4), BAT-specific genes were measured in 3T3-L1 cells treated with each agent. BMP4 stimulates WAT-to-BAT differentiation in preadipocytes, so it was used as a positive control [34]. Twenty nanograms of BMP4 significantly increased expression of PPARα, PPARδ, PRDM16, PGC1α and UCP1, as did 25 μM GN (Figure 5A,B). Co-treatment with both agents appeared to have an additive effect (Figure 5A,B). In fact, GN upregulation of PPARδ, PGC1α and UCP1 was greater than that of BMP4 (Figure 5A,B). PPARα and PPARδ are associated with increased mitochondrial FAO and upregulate expression of thermogenic genes, including PRDM16, PGC1α and UCP1 [35]. In addition, GN and BMP4 increased mRNA levels of thermogenic -genes, including cell death-inducing DNA fragmentation factor alpha-like effector A (*Cidea*), cytochrome c oxidase subunit VIIa polypeptide (*Cox7a1*), transmembrane protein 26 (*Tmem26*) and the BAT-specific genes *Prdm16*, *Pgc1α* and *Ucp1* (Figure 5C). Moreover, GN and BMP4 co-treatment synergistically increased expression of *Prdm16*, *Pgc1α* and *Ucp1* (Figure 5C). Additionally, both GN and BMP4 increased CO_2_ production in 3T3-L1 cells; co-treatment had an additive effect (Figure 5D) Especially, the effect of GN on CO_2_ production was greater than that of BMP4. Taken together, these findings indicated that GN induced adipose browning by elevating expression of the thermogenic factor UCP1, increasing energy expenditure in 3T3-L1 cells. 

Finally, we investigated the relative effects of GN and BMP4 on regulation of lipid metabolism. Prior reports have demonstrated that BMP4 stimulates WAT-to-BAT differentiation in mature adipocytes by increasing AMPK activation [36]. Treatment with BMP4 significantly up-regulated p-AMPK and p-ACC in 3T3-L1 cells, and GN/BMP4 co-treatment synergistically increased p-AMPK and p-ACC (Figure 6A,B). Transcription factors associated with adipogenesis, including PPARγ and C/EBPα, were measured. BMP4 increased PPARγ expression and decreased C/EBPα expression (Figure 6C). PGC1α is a key regulatory factor in BMP4-mediated adipocyte browning, such that PPARγ associated with PGC1α could increase during adipocyte differentiation [34]. However, GN treatment effectively reduced the adipogenic markers PPARγ and C/EBPα in 3T3-L1 cells. Together, these findings suggested that GN induction of a BAT-like phenotype during 3T3-L1 differentiation was greater than that of BMP4, a well-known inducer of adipocyte browning. 

## 3. Discussion

Obesity occurs due to an imbalance between energy intake and energy expenditure [37]. WAT accumulates excessive energy in the form of TGs, which is directly associated with obesity. Conversely, BAT dissipates energy as heat, suggesting that it can alleviate obesity and obesity-associated metabolic disease [5]. Thus, promoting a BAT-like phenotype in WAT, known as WAT browning, could increase energy dissipation. A variety of dietary molecules, such as berberine, ginsenoside Rg1 and silk peptide, are reported to modulate adipocyte expression of thermogenic transcription factors [37,38]. In the present study, we aimed to investigate the anti-obesity effects of GN in the context of AMPK-mediated lipid metabolism and adipocyte browning.

GN is a major lignan derived from SC, which has been widely used as a traditional herbal medicine in Asian countries. GN possesses known neuroprotective, hepatoprotective and anti-cancer properties [39,40,41]. A recent study suggested that SC suppresses obesity in high-fat diet-induced obese rats and 3T3-L1 cells by decreasing expression of C/EBPα and PPARγ. They reported that 100 μM GN ameliorates early stages of adipocyte differentiation rather than post stage with decreasing adipogenic gene expression [42]. However, the effects of individual SC components on lipogenesis, FAO and adipose browning have not yet been determined. Therefore, in the present study, we assessed the effects of GN, an important component of SC, using an in vitro model. 

Adipogenesis involves the development of fully differentiated mature adipocytes from pre-adipocytes. This mechanism is primarily regulated by PPARγ, C/EBPα, FAS and FABP4, which are crucial for the late stages of adipocyte differentiation [43]. In present study, we identified that GN inhibited adipogenesis by down-regulating expression of these adipocyte-specific transcription factors in 3T3-L1 cells. Indeed, ORO data indicates that GN effectively reduced lipid accumulation over 8 days of differentiation compared with control cells. We further determined whether decreased lipid accumulation was associated with disruption of TG synthesis. GN significantly reduced intracellular TGs in 3T3-L1 cells by decreasing expression of lipogenic proteins, including SREBP1, LPAATθ, DGAT1 and Lipin1. Lipolysis is the hydrolysis process of TG into glycerol and free fatty acids in adipocytes [44]. p-HSL, the key enzyme that performs in TG lipolysis, was elevated by GN treatment in 3T3-L1. This result is closely correlated with the effect of GN on the FAO gene expression by providing free fatty acids. Thus, our findings suggested that GN regulated lipid metabolism by downregulating adipogenesis and TG synthetic genes, suggesting that it can suppress obesity through WAT browning. 

UCP1 is a mitochondrial inner membrane protein and dissipates energy in activated BAT or BAT-like WAT cells. FGF21 also plays a role in energy metabolism [45]. FGF21 is preliminary expressed in adipose tissue, skeletal muscle or liver. Recent study reported that FGF21 increases energy expenditure by stimulating PGC1α and UCP1 expression, which are important for mitochondrial biogenesis in adipocytes [46]. In the present study, western blotting revealed that GN increased expression of the thermogenic genes FGF21, PRDM16, PGC1α and UCP1 in 3T3-L1 cells in a dose-dependent manner. PRDM16 and PGC1α are key drivers of WAT browning and interact with transcription factors of UCP1 [47]. Immunofluorescence revealed increased intensity of MitoTracker and UCP1 in GN-treated cells. These mitochondrial biogenesis activity is closely associated with energy metabolism by increasing increases oxygen consumption rate and β-oxidation in differentiated cells [48,49]. In addition, GN elevated the mRNA expression of *Pparα*, *Pparδ*, *Cpt1* and *Cpt2*, which are vital for mitochondrial biogenesis and FAO. These genes facilitate conversion of fatty acids into coenzyme-A ester and modulate transport of fatty acids across the mitochondrial membrane for FAO. Moreover, FAO genes contribute to mitochondrial activity and UCP1 activation [38]. Together, these findings indicated that GN promoted 3T3-L1 browning by increasing expression of UCP1 and FAO genes and mitochondrial biogenetic activity.

In the present study, we evaluated and compared the browning effect of GN with that of BPM4, a known regulator of adipose browning. BMP4 is a member of the bone morphogenic protein family and regulates commitment to the adipocyte lineage. Recent studies have demonstrated that BMP plays important roles in terminal adipogenesis and the metabolic processes of adipocytes. Specifically, BMP4 overexpression in WAT induces a BAT-like phenotype, including decreased fat droplets and adipose size and increased mitochondrial activity and FAO [34,50]. In the present study, we identified that BMP4 significantly up-regulated FAO genes, mitochondrial biogenesis genes and thermogenic genes to a similar extent as that of GN, although the effect of GN was greater than that of BMP4 in some cases. Together, these data demonstrated that GN treatment enhances UCP1 expression in 3T3-L1 cells, increasing energy expenditure. Additionally, immunofluorescence revealed that GN increased mitochondrial activity, concomitant with increased UCP1 expression. Lastly, CO_2_ production in GN-treated adipocytes was significantly greater than that of control or BMP4-treated cells. These data suggest that GN could increase energy expenditure by upregulating thermogenic genes in 3T3-L1 cells. Taken together, the results suggest that GN regulates adipocyte browning by upregulating UCP1 expression and increasing energy expenditure in adipocytes.

AMPK is a known master regulator of whole-body metabolism. AMPK also regulates mitochondrial biogenesis by activating transcription of UCP1, a key regulator of BAT differentiation [51]. A recent study reported that gomisin component inhibits hepatic lipogenesis via the AMPK pathway in HepG2 cells [20]. In our study, GN treatment dose-dependently increased AMPK phosphorylation in 3T3-L1 cells. Therefore, we investigated whether AMPK activation was necessary for GN induction of thermogenesis and reduced lipid accumulation in adipocytes. Cells were treated with the AMPK activator AICAR or AMPK inhibitor CC (Compound C, 6-[4-(2-Piperidin-1-yl-ethoxy)-phenyl)]-3-pyridin-4-yl-pyrrazolo[1,5-a]-pyrimidine) with or without GN. GN treatment increased phosphorylated AMPK, phosphorylated ACC and UCP1 expression. These data indicated that the thermogenic effects of GN were mediated by AMPK activation. Also, GN activated AMPK as much as did BMP4. Consequently, ACC phosphorylation and expression of thermogenic factors such as UCP1 were also increased in GN-treated cells relative to BMP4-treated cells. Taken together, our data suggested that GN stimulated WAT-to-BAT differentiation by activating AMPK signaling to regulate lipid and energy metabolism.

In conclusion, GN induced WAT-to-BAT differentiation of 3T3-L1 adipocytes by increasing expression of FAO and thermogenic genes. Moreover, GN ameliorated TG accumulation by reducing expression of adipogenesis and lipogenesis genes. Also, the browning effect of GN on adipocytes was mediated by AMPK activation. Our findings therefore suggest that GN is a potent regulator of adipocyte browning and lipid metabolism, so it may have therapeutic potential in the treatment of obesity and obesity-associated metabolic disease. 

## 4. Materials and Methods

### 4.1. Materials 

Gomisin N (≥98% purity, C_23_H_28_O_6,_ GN) was purchased from YuanYe Biotechnology (Shanghai, China). Dulbecco’s modified Eagle’s medium (DMEM), bovine calf serum (BCS), fetal bovine serum (FBS), penicillin–streptomycin (P/S), recombinant human BMP4, trypsin and ethylenediaminetetraacetic acid (EDTA) were purchased from Gibco (Gaithersburg, MD, USA). Dexamethasone (DEX), 3-isobutyl-1-methylxanthine (IBMX), insulin, Oil Red O (ORO), dimethyl sulfoxide (DMSO), phosphatase inhibitor cocktails I and II, AICAR and CC were purchased from Sigma-Aldrich (St. Louis, MO, USA). Phosphate-buffered saline (PBS) was purchased from iNtRON Biotechnology (Gyeonggi, South Korea). Antibodies against C/EBPα (sc-61), FAS (sc-20140), SREBP1c (sc-366), LPAATθ (sc-514164), DGAT1 (sc-32861), Lipin1 (sc-98450), PPARα (sc9000), PPARγ (sc7196), PGC1α (sc13067), CPT1 (sc393070), UCP1 (sc6529) and GAPDH (sc365062) were purchased from Santa Cruz Biotechnology, Inc. (Santa Cruz, CA, USA). Antibodies against p-ACC (cs3661, Ser79) p-AMPK (cs2535, Thr172), AMPK (cs2603), p-HSL(cs4139) and ACC (cs3662) were purchased from Cell Signaling Technologies (Danvers, MA, USA). Antibodies against PPARδ (ab23673), PRDM16 (ab202344), UCP1 (ab23841), FGF21 (ab64857) and CPT1 (ab1285568) were purchased from Abcam (Cambridge, UK). 

### 4.2. Cell Viability Assay 

The effect of GN on 3T3-L1 cell viability was assessed by the XTT assay. Cell viability was measured using a cell proliferation assay kit according to the manufacturer’s protocol (WelCount TR005-01, WelGENE, Seoul, Korea). Briefly, adipocytes were treated with GN in 96 well plates and incubated for 24 h. Subsequently, 2,3-bis(2-methoxy-4-nitro-5-sulfophenyl)-2H-tetrazolium-5-carboxanilide (XTT) reagent and N-methyl dibenzopyrazine methyl sulfate (PMS) were added to each well for 4 h. Absorbance was measured at wavelengths of 450 and 690 nm using a plate reader (BioTek Instruments, Inc. Winooski, VT, USA).

### 4.3. Cell Culture

3T3-L1 pre-adipocytes (CL-173; American Type Culture Collection, Manassas, VA, USA) were grown at 37 °C in a humidified 5% CO_2_ incubator in DMEM medium containing 10% bovine calf serum, 1% P/S and 3.7 g/L sodium bicarbonate. Cells were cultured to 100% confluence by replacing growth medium every 2 days. After a further culture for 2 days, this medium was replaced with DMEM containing 10% fetal bovine serum (FBS), 1% P/S and MDI (0.5 mM 3-isobutyl-1-methylxanthine, 1 μM dexamethasone and 4 μg/mL insulin). Initial differentiation of confluent adipocytes was induced by two days treatment with DMEM differentiation medium containing 10% FBS, 1% P/S and 5 μg/mL insulin. Culture medium was then refreshed every 2 days with DMEM maintenance medium supplemented with 10% FBS and 5 μg/mL insulin. A 50 mM GN stock solution was prepared in dimethyl sulfoxide and diluted with differentiation medium to 6.25, 12.5 and 25 µM. To induce 3T3-L1 browning, cells were incubated in differentiation medium containing MDI and 20 ng/mL BMP4. Post-confluent cells were pretreated with 10 µM AICAR or 10 µM CC for 4 h, after which cells were differentiated in the presence or absence of GN for 24 h. 

### 4.4. Oil Red O Staining 

After 8 days of differentiation, cells were fixed in 4% formaldehyde for 1 h. Fixed adipocytes were stained with 0.5% ORO solution in 6:4 (*v*/*v*) isopropanol: distilled water for 1 h, and subsequently washed twice with distilled water. Stained cells were dried, imaged and eluted with 100% isopropanol. The absorbance of these eluents was determined at 490 nm using a plate reader (BioTek Instruments, Inc.).

### 4.5. Measurement of Triacylglycerols (TGs)

Eight-day differentiated 3T3-L1 cells were harvested in lysis buffer containing 1% Triton-100, 150 mM NaCl, 4 mM EDTA, 20 mM Tris HCl (pH 7.4) and protease inhibitor cocktail and lysed completely using a horn-type sonicator. TG, an index of lipid accumulation, was quantitatively measured using a commercially available TG assay kit (ZenBio, Research Triangle Park, NC, USA) according to the manufacturer’s instructions. The absorbance of these eluents was determined at 540 nm using a plate reader (BioTek Instruments, Inc.).

### 4.6. Measurement of Carbon Dioxide (CO_2_)

Carbon dioxide production was measured using a carbon dioxide assay kit according to the manufacturer’s protocol (Crystal Chem, Downers, IL, USA). Absorbance was measured using a PowerWave HT microplate spectrophotometer (BioTek Instruments, Inc.) at 405 nm.

### 4.7. Quantitative Real-Time Polymerase Chain Reaction (qPCR)

Total RNA from mature adipocytes was extracted using TRIzol reagent (Invitrogen, Carlsbad, CA, USA) according to the manufacturer’s protocol. cDNA was synthesized from 1 μg total RNA using a Maxime RT PreMix Kit (iNtRON Biotechnology, Gyeonggi, South Korea). To quantify gene expression levels, cDNA was amplified using a SYBR Green 2X master mix Kit (Roche Diagnostics GmbH, Mannheim, Germany) and Bio-Rad CFX96 Real-Time Detection System (BioRad, Hercules, CA, USA). Experiments were conducted in quadruplicate for each sample, and relative mRNA levels were normalized against 18S rRNA. mRNA levels were calculated as a ratio, using the 2^−∆∆CT^ method for comparing relative mRNA expression between groups. Oligonucleotide primers are listed in Table 1.

### 4.8. Western Blot Analysis 

Cells were lysed using lysis buffer (iNtRON Biotechnology, Seoul, Korea) with protease inhibitor cocktail (Biomake B14001, Houston, TX, USA), phosphatase inhibitor 2 (Sigma P5726) and phosphatase inhibitor 3 (Sigma P0044). Lysate proteins were quantified using a protein assay kit (Bio-Rad, Hercules, CA, USA). Equal amounts of protein (20 μg) were subjected to sodium dodecyl sulfate polyacrylamide gel electrophoresis (SDS-PAGE) and electrotransferred to polyvinylidene fluoride (PVDF) membranes.

Membranes were blocked in 5% skimmed milk for 1 h and washed with Tris-buffered saline with Tween 20 (TBS-T). Membranes were immunoblotted with primary antibodies (1:1000) overnight at 4 °C and washed with TBS-T buffer. Subsequently, membranes were incubated with secondary antibodies conjugated to horseradish peroxidase (1:2000) at room temperature for 6 h and washed with TBS-T buffer. Bands were detected using LAS imaging software (Fuji, New York, NY, USA).

### 4.9. Statistical Analysis

Each experiment was performed at least 3–4 times. Results are expressed as mean ± standard deviation (SD). A one-way ANOVA with Duncan’s test (SPSS, 17.0, Chicago, IL, USA) was used to analyze differences among multiple groups. *p* < 0.05 was considered statistically significant. mRNA data with two groups were analyzed using a Student’s *t*-test (SPSS) and expressed as mean ± SD. Values with different letters are significantly different, *^*^ p* < 0.05, *^**^ p* < 0.01, *^***^ p* < 0.001.

## Figures and Tables

**Figure 1 ijms-21-02153-f001:**
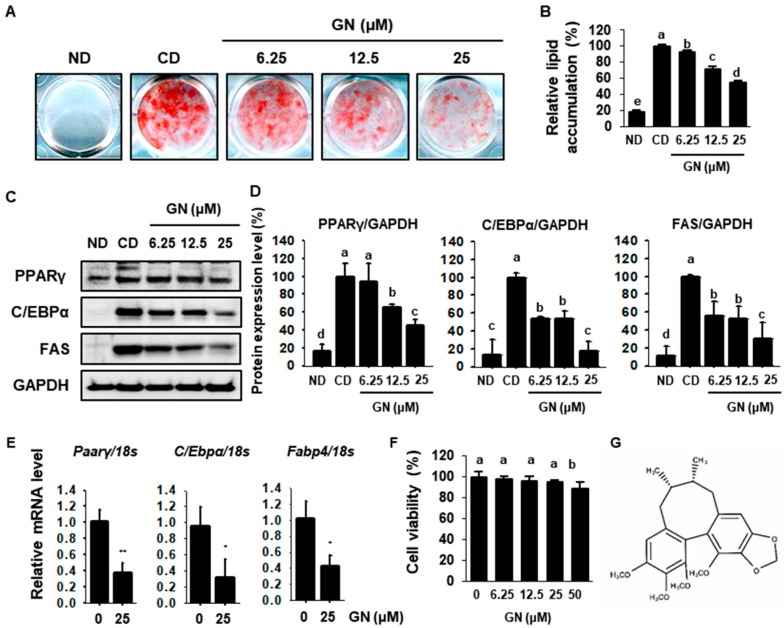
Effects of Gomisin N (GN) on lipid accumulation in 3T3-L1 cells. (**A**,**B**) Oil red O staining images of 3T3-L1 cells treated with GN for 8 days of differentiation and quantification of Oil red O levels. (**C**,**D**) Western blotting for the adipogenic proteins peroxisome proliferator-activated receptor gamma (PPARγ), CCAAT enhancer-binding protein alpha (C/EBPα) and fatty acid synthase (FAS) in 3T3-L1 cells and band density quantification. (**E**) Relative mRNA expression levels of the adipogenic genes *Pparγ*, *C/Ebpα* and *ap2* were determined by qPCR and normalized to 18S rRNA expression. (**F**) Cell viability of 3T3-L1 cells treated with GN for 24 h, as measured by the XTT assay. (**G**) Chemical structure of GN. Protein data are expressed as mean ± SD (*n* = 4). Values with different letters are significantly different, *p* < 0.05 (a > b > c > d > e). mRNA data are expressed as mean ± SD (*n* = 4) and values with different letters indicating significant differences, *^*^ p* < 0.05, *^**^ p* < 0.01. ND; non-differentiation. CD; control-differentiation.

**Figure 2 ijms-21-02153-f002:**
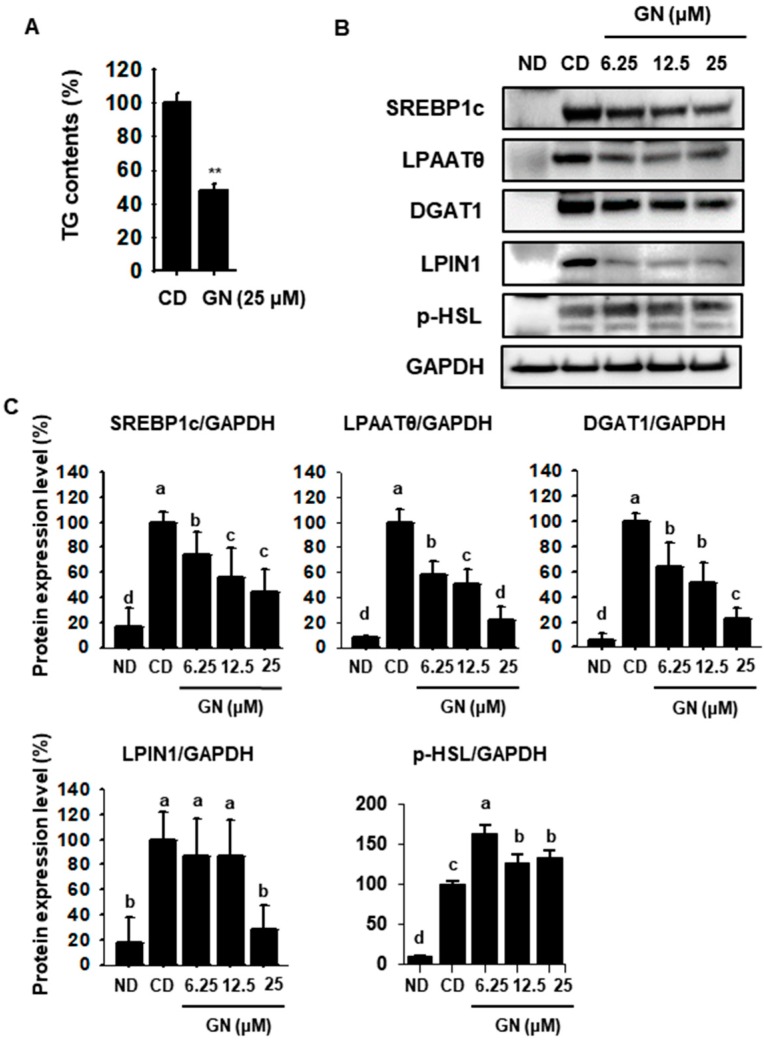
Effects of GN on triglyceride synthesis in 3T3-L1 cells. (**A**) Intracellular triglycerides (TGs) in 3T3-L1 cells treated with GN were measured using a TG ELISA kit. (**B**,**C**) Western blotting for the lipogenesis factors sterol regulatory element-binding transcription factor 1c (SREBP1c), lysophosphatidic acid acyltransferase theta (LPAATθ), diacylglycerol acyltransferase 1 (DGAT1), phosphatidate phosphatase (LPIN1) and phosphorylated hormone sensitive lipase (p-HSL) in 3T3-L1 cells and quantification of band density. Protein data are expressed as mean ± SD (*n* = 4). Values with different letters are significantly different, *p* < 0.05 (a > b > c > d).

**Figure 3 ijms-21-02153-f003:**
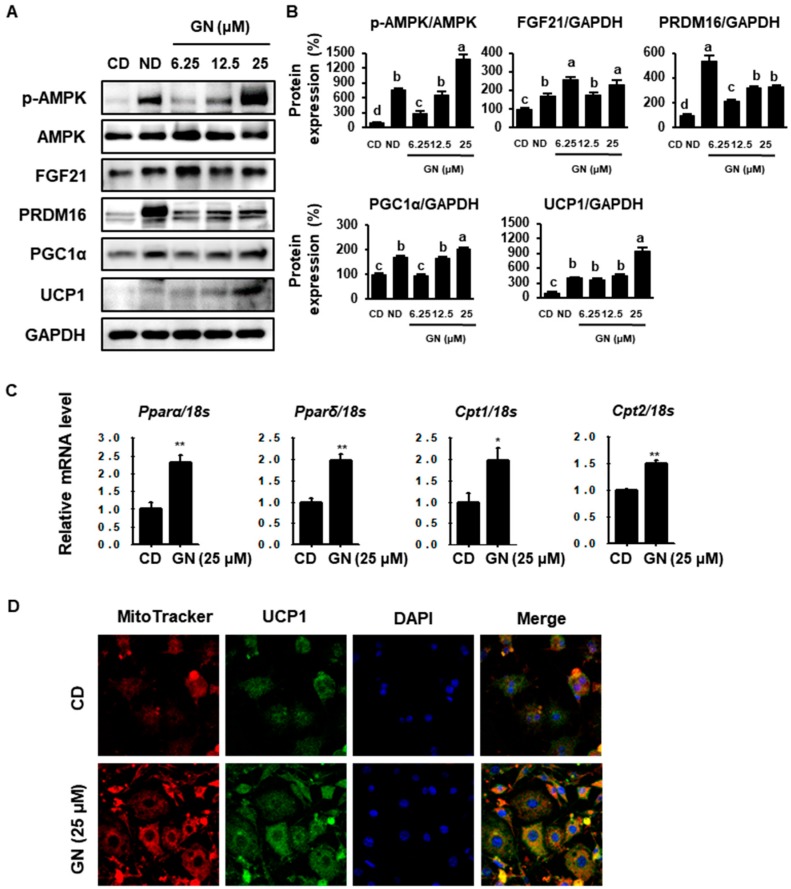
Effects of GN on expression of thermogenic markers and fatty acid oxidation-related genes. (**A**,**B**) Western blotting for the lipogenesis factors phosphorylated AMP-activated protein kinase (p-AMPK), AMPK, fibroblast growth factor (FGF21), PR domain containing 16 (PRDM16), peroxisome proliferator-activated receptor gamma co-activator 1 alpha (PGC1α) and uncoupling protein 1(UCP1) in 3T3-L1 cells and quantification of band density. (**C**) Relative mRNA expression levels of the adipogenic genes *Pparα*, *Pparδ*, *Cpt1* and *Cpt2* were determined by qPCR and normalized to 18S rRNA expression. (**D**) Immunofluorescence image of 3T3-L1 (400× magnification). Protein data are expressed as mean ± SD (*n* = 4). Values with different letters are significantly different, *p* < 0.05 (a > b > c > d). mRNA data are expressed as mean ± SD (*n* = 4). * *p* < 0.05, *^**^ p* < 0.01.

**Figure 4 ijms-21-02153-f004:**
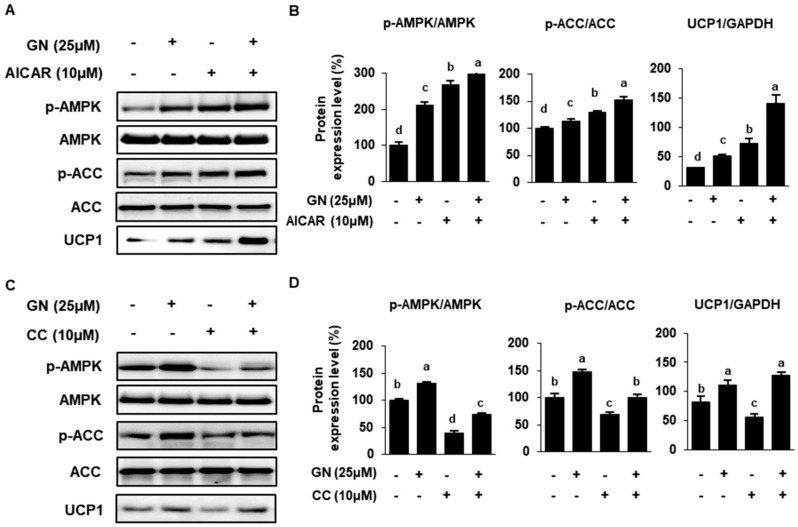
Effects of AMPK activation in 3T3-L1 cells treated with GN. (**A**,**B**) 3T3-L1 cells were treated with 10 μM AICAR ± 25 μM GN, and effects on protein phosphorylation and/or expression were determined by Western blotting. (**C**,**D**) 3T3-L1 cells were treated with 10 μM CC ± 25 μM GN, and effects on protein phosphorylation and/or expression were determined by Western blotting. Protein data are expressed as mean ± SD (*n* = 4). Values with different letters are significantly different, *p* < 0.05 (a > b > c > d). AICAR; 5-aminoimidazole-4-carboxamide ribonucleotide, CC; Compound C, 6-[4-(2-Piperidin-1-yl-ethoxy)-phenyl)]-3-pyridin-4-yl-pyrrazolo[1,5-a]-pyrimidine.

**Figure 5 ijms-21-02153-f005:**
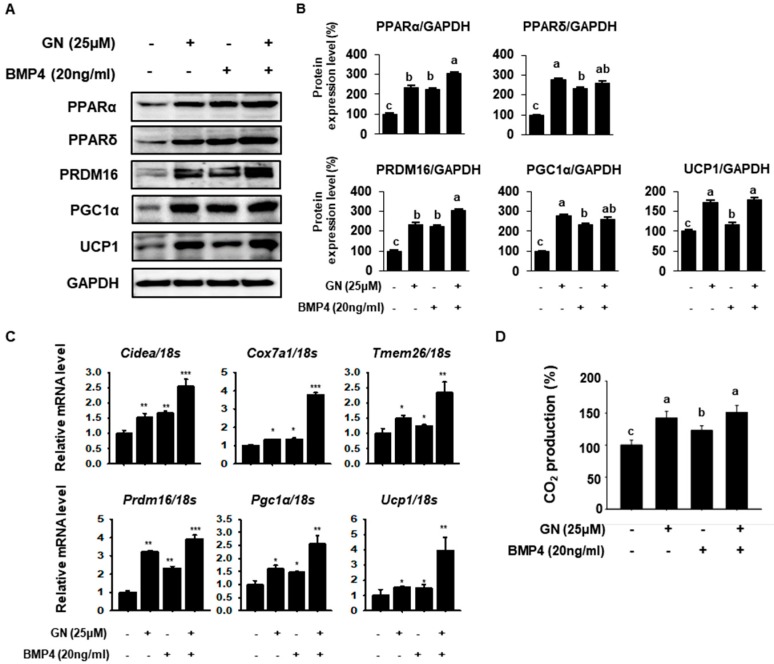
Synergistic effects of BMP4 and GN on thermogenesis in 3T3-L1 cells. (**A**,**B**) Western blotting for the browning factors PPARα, PPARδ, PRDM16, PGC1α and UCP1 in 3T3-L1 cells and band density quantification. All data are expressed as mean ± SD (*n* = 4) and values with different letters are significantly different, *p* < 0.05 (a > b > c > d). (**C**) Relative mRNA expression levels of the thermogenic genes *Cidea*, *Cox7a1*, *Tmem26*, *Prdm16*, *Pgc1α* and *Ucp1* were determined by qPCR and normalized to 18S rRNA expression. (**D**) Relative CO_2_ production was analyzed using a carbon dioxide assay kit according to the manufacturer’s instructions. mRNA data are expressed as mean ± SD (*n* = 4) and values with different letters are significantly different *^*^ p* < 0.05, *^**^ p* < 0.01, *^***^ p* < 0.001. BMP4; Bone morphogenetic protein 4.

**Figure 6 ijms-21-02153-f006:**
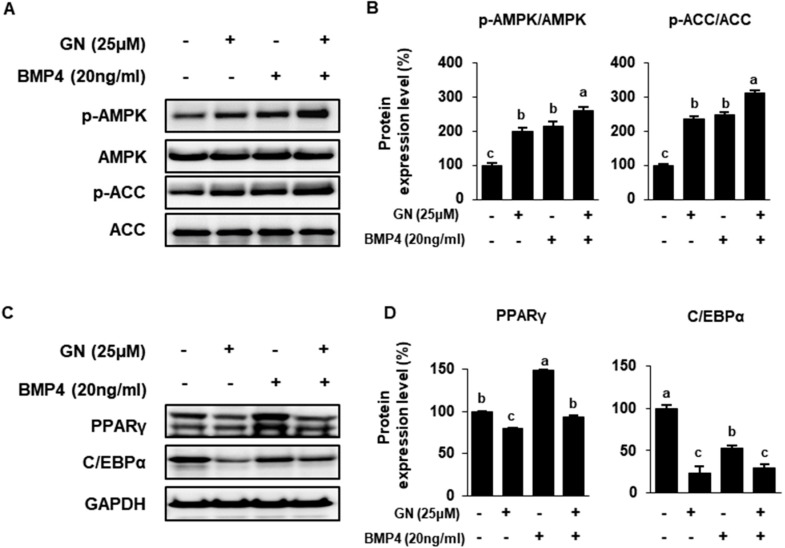
Additive effects of BMP4 and GN on AMPK activation and adipogenesis in 3T3-L1 cells. 3T3-L1 cells were treated with 20 ng/mL BMP4 ± 25 μM GN, h-and effects on protein phosphorylation and/or expression were determined. (**A**,**B**) Western blotting for p-AMPK, AMPK, phosephorylated acetyl-CoA carboxylase (p-ACC) and ACC in 3T3-L1 cells and quantification of band density. (**C**,**D**) Western blotting for the adipogenic proteins PPARγ and C/EBPα in 3T3-L1 cells and band density quantification. Protein data are expressed as mean ± SD (*n* = 4). Values with different letters are significantly different, *p* < 0.05 (a > b > c > d). BMP4; Bone morphogenetic protein 4.

**Table 1 ijms-21-02153-t001:** Primer sequences used for qPCR analysis.

Gene	Forward Primer	Reverse Primer
*18S*	GCAATTATTCCCCATGAAC	GGCCTCACTAAACCATCCAA
*Cidea*	TGCTCTTCTGTATCGCCCAGT	GCCGTGTTAAGGAATCTGCTG
*Cox7a1*	CAGCGTCATGGTCAGTCTGT	AGAAAACCGTGTGGCAGAGA
*Tmem26*	CCCTACTCTGGTCTCTGGCA	GGAAGGGACCGTCTTGGATG
*Prdm16*	CAGCACGGTGAAGCCATTC	GCGTGCATCCGCTTGTG
*Pgc1α*	CCCTGCCATTGTTAAGACC	TGCTGCTGTTCCTGTTTTC
*Ucp1*	ACTGCCACACCTCCAGTCATT	CTTTGCCTCACTCAGGATTGG

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
