# Peer review of "Gomisin N from Schisandra chinensis Ameliorates Lipid Accumulation and Induces a Brown Fat-Like Phenotype through AMP-Activated Protein Kinase in 3T3-L1 Adipocytes"

_ijms, 2020, doi:10.3390/ijms21062153_

Round 1

Reviewer 1 Report

The manuscript is well written, investigating activation of BAT as a therapeutical intervention. The authors display good knowledge of adipocytes, including adipogenesis and beiginng.

The authors suggest that there is an increase in lipolysis; this should be measured via glycerol content in media and western blots for markers such as HSL phosphorylation.

Can GN initiate similar effects in cells that are already 8-days post diff rather than supplemented throughout differentiation.

Can the authors provide a real time measure of metabolic parameters, such as oxygen consumption using a seahorse, Clarke electrodes, or oroboros.

Measuring mitochondrial content in treated versus control as a further beneficial measurement. In addition other readings of mitochondria such as dynamics, oxphos proteins.

Can the authors comment on TZD benefits which act on PPARg activation in adipose versus GN PPARg inhibition. Clinically TZDs and PPARg activation have been shown to be beneficial, in particular improving insulin sensitivity.

Author Response

Dear reviewer,

Thank you for considering our manuscript for publication in International Journal of Molecular Sciences. We are very pleasure to have been given the opportunity to revise our manuscript, “Gomisin N from Schisandra chinensis ameliorates lipid accumulation and induces a brown fat-like phenotype through AMP-Activated Protein Kinase in 3T3-L1 adipocytes”. We have addressed the reviewer’s comments point-by-point and made the necessary changes to the manuscript. Please see the attachment.

We hope that the manuscript is acceptable for publication in International Journal of Molecular Sciences. We appreciate again to consider this paper for publication in IJMS, and declare that authors of this work have no conflict of interests.

Sincerely, yours,

Boo-Yong Lee, Ph.D.

Reviewer 1

1) The authors suggest that there is an increase in lipolysis; this should be measured via glycerol content in media and western blots for markers such as HSL phosphorylation.

 Answer: We appreciated for your suggestion. As reviewer mentioned, signaling pathways of lipogenesis and lipolysis are must be included to explain TG reducing effect. Therefore, we determined p-HSL expression in 3T3-L1 treated with GN and presented in Figure 2. This is described in the Results 2.2. and Discussion.; “In other hands, GN effectively increased the phosphorylated hormone sensitive lipase (p-HSL) expression, which catalyzes the hydrolysis of cellular triacylglycerol and diacylglycerol in adipocytes.”-in result 2.2. ““Lipolysis is the hydrolysis process of TG into glycerol and free fatty acid in adipocytes [1]. p-HSL, the key enzyme that performs in TG lipolysis, was elevated by GN treatment in 3T3-L1. This result is closely correlated with the effect of GN on the FAO gene expression by providing free fatty acids .”–in Disucussion.

2) Can GN initiate similar effects in cells that are already 8-days post diff rather than supplemented throughout differentiation.

 Answer: We appreciate for your kind advice. We referred and designed a study in reference 28 (Jang et al., 2017) [2], which is characterized the effect of GN on 3T3-L1 cells (reduced PPARγ, C/EBPα and FAS, reduced lipid accumulation, cell viability). In reference 28, Gomisin N obtained from ChemFaces did not affect the cell viability in 3T3-L1 up to high concentration of 10-100 µM, and reduced lipid accumulation by decreasing adipogenic gene expression. Moreover, they reported that GN ameliorates early stages of adipocyte differentiation rather than post stage. In our study, because GN showed cell toxicity at 50 µM which is a lower concentration than that of previous study, thereby cell treated with 6.25-25 µM of GN throughout differentiation duration.

3) Can the authors provide a real time measure of metabolic parameters, such as oxygen consumption using a seahorse, Clarke electrodes, or oroboros.

 Answer: We appreciated for your suggestion. We agree that energy expenditure have to include carbon dioxide emission and oxygen consumption. At the beginning of research, we wanted to measure oxygen consumption rate (OCR) in cells treated GN by using seahorse XF analyzer and fully considered this. However, because we didn’t have metabolic measurement analyzer, we could not analyze the metabolic change including oxygen consumption using it due to financial and technical problem. As a laboratory member for browning research, we have tried to prove the effect of dietary materials on energy expenditure and please understand that there are these limitations. Instead, we have suggested the mitochondrial activity by using mitotracker-red immunofluorescence analysis. Mitochondrial biogenesis activity is closely associated with energy metabolism in adipocytes. Regarding on it, we descripted as “These mitochondrial biogenesis activity is closely associated with energy metabolism by increasing increases oxygen consumption rate and β-oxidation in differentiated cells” in Discussion. [3-5]

4) Measuring mitochondrial content in treated versus control as a further beneficial measurement. In addition other readings of mitochondria such as dynamics, oxphos proteins.

 Answer: We appreciate for your kind advice. As reviewer mentioned, recent study reported that mitochondrial remodeling is involved the quantitative and qualitative change of the mitochondria during browning [4,6]. In this regard, as we answered in Question 3, GN stimulated mitochondrial biogenesis by increasing thermogenic genes PGC1α, PRDM16 and UCP1 during WAT-to-BAT differentiation. Even though we tried to measure mitochondrial contents using transmission electron microscope during experimental period, we could not analyze this due to our technical problem, unfortunately. However, our immunofluorescence data using mitotracker red shows not only but qualitative change (increased intensity) also quantitative change (increased number of cells which have wide range intensity).

5) Can the authors comment on TZD benefits which act on PPARg activation in adipose versus GN PPARg inhibition. Clinically TZDs and PPARg activation have been shown to be beneficial, in particular improving insulin sensitivity.

 Answer: As reviewer mentioned, PPARγ is one of thiazolidinedione (TZD) and is important to energy metabolism by rebulating insulin sensitivity [7]. Increasing insulin sensitivity by PPARγ upregulation is occurred when PPARγ lead to increase fat formation during adipocyte differentiation. However, TZD is reported to increase insulin sensitivity in type 2 diatetes patients who correlated with excessive fat accumulation [8]. In this regard, our findings showed that GN reduces PPARγ expression during 8 days differentiation, while GLUT4 expression has no significant difference (Please see attached image). Therefore, PPARγ is thoght to increase to stimulate PGC1α and then decrease again in intermediated differentiation in 3T3-L1 treated with GN.

Reference

  1. Okazaki, H.; Osuga, J.-i.; Tamura, Y.; Yahagi, N.; Tomita, S.; Shionoiri, F.; Iizuka, Y.; Ohashi, K.; Harada, K.; Kimura, S. Lipolysis in the absence of hormone-sensitive lipase: evidence for a common mechanism regulating distinct lipases. Diabetes 2002, 51, 3368-3375.
  2. Park, H.J.; Cho, J.-Y.; Kim, M.K.; Koh, P.-O.; Cho, K.-W.; Kim, C.H.; Lee, K.-S.; Chung, B.Y.; Kim, G.-S.; Cho, J.-H. Anti-obesity effect of Schisandra chinensis in 3T3-L1 cells and high fat diet-induced obese rats. Food Chemistry 2012, 134, 227-234.
  3. Si, Y.; Palani, S.; Jayaraman, A.; Lee, K. Effects of forced uncoupling protein 1 expression in 3T3-L1 cells on mitochondrial function and lipid metabolism. Journal of lipid research 2007, 48, 826-836.
  4. Lee, J.H.; Park, A.; Oh, K.-J.; Lee, S.C.; Kim, W.K.; Bae, K.-H. The Role of Adipose Tissue Mitochondria: Regulation of Mitochondrial Function for the Treatment of Metabolic Diseases. International journal of molecular sciences 2019, 20, 4924.
  5. Kusminski, C.M.; Scherer, P.E. Mitochondrial dysfunction in white adipose tissue. Trends in endocrinology & metabolism 2012, 23, 435-443.
  6. Wood dos Santos, T.; Cristina Pereira, Q.; Teixeira, L.; Gambero, A.; A Villena, J.; Lima Ribeiro, M. Effects of polyphenols on thermogenesis and mitochondrial biogenesis. International journal of molecular sciences 2018, 19, 2757.
  7. Hauner, H. The mode of action of thiazolidinediones. Diabetes/metabolism research and reviews 2002, 18, S10-S15.
  8. Soccio, R.E.; Chen, E.R.; Lazar, M.A. Thiazolidinediones and the promise of insulin sensitization in type 2 diabetes. Cell metabolism 2014, 20, 573-591.

Reviewer 2 Report

To the authors

Firstly, may I congratulate the authors on a clear manuscript. The effects of Gomisin N (GN) on lipid accumulation and ‘browning’ of adipocytes are interesting. However, I would be interested to hear the authors rationale for the studies – how do these results extend our understanding of GN biology beyond what is already known? For example, Jang et al., 2017 (reference 28) characterised the effect of GN on 3T3-L1 cells (reduced PPARᵧ, C/EBPα and FAS, reduced lipid accumulation, cell viability) whilst demonstrating an effect in vivo. Whilst there is novelty here, could this be further extended? Why did the authors choose the 3T3-L1 cell line? Did the authors consider confirming their findings in primary adipocytes? The authors measured CO2 production in 3T3-L1 cells in response to treatment with GN and suggest that energy expenditure is increased in response to treatment. Did the authors consider using a Seahorse XF Analyser to confirm effects on the oxygen consumption rate? A robust response to these questions would be welcomed.

Whilst my remaining comments are rather trivial in nature, and I have nevertheless divided them into major and minor.

Major comments

As a reader, I would like some indication of what GN is in the title.

Given that GN suppresses cell proliferation and cell cycle progression in the early stages of adipogenesis how do the authors account for this in their study? This needs to be further highlighted in the results/methods.

Given the effects of treatment I would be interested to see the effect on FGF21 - was this measured? If so/possible, please report.

Did the authors investigate how GN activates AMPK?

Minor comments

The authors refer to an XTT assay in line 96 – had this been previously defined?

Could the authors add a table of antibodies used in Western blotting studies (containing details of concentration and supplier)?

The UCP1 (and PGC1α) blots in latter figures are highly variable – please can the authors account for this? For example, in figure 3 the UCP1 blot is ‘dirty’, whilst is relatively ‘clean’ in figure 4. Yet in figure 5, multiple bands are detected?

Author Response

Dear reviewer,

Thank you for considering our manuscript for publication in International Journal of Molecular Sciences. We are very pleasure to have been given the opportunity to revise our manuscript, “Gomisin N from Schisandra chinensis ameliorates lipid accumulation and induces a brown fat-like phenotype through AMP-Activated Protein Kinase in 3T3-L1 adipocytes”. We have addressed the reviewer’s comments point-by-point and made the necessary changes to the manuscript. Please see the attachment.

We hope that the manuscript is acceptable for publication in International Journal of Molecular Sciences. We appreciate again to consider this paper for publication in IJMS, and declare that authors of this work have no conflict of interests.

Sincerely, yours,

Boo-Yong Lee, Ph.D.

Reviewer 2

1) How do these results extend our understanding of GN biology beyond what is already known? For example, Jang et al., 2017 (reference 28) characterised the effect of GN on 3T3-L1 cells (reduced PPARᵧ, C/EBPα and FAS, reduced lipid accumulation, cell viability) whilst demonstrating an effect in vivo. Whilst there is novelty here, could this be further extended?

 Answer: We appreciated for your suggestion. Although previous study (reference 28) investigated effect of GN on adipogenesis and cell viability, browning effect was not explored. Existing obesity studies such as reference28 have been a mechanism to suppress fat accumulation in white fat. However, we have highlighted not only signaling pathway of TG synthesis inhibition but also thermogenic mechanisms including PRDM16, PGC1 and UCP1 gene expression by treatment GN in 3T3-L1. Our study proposes a new anti-obesity strategy for brown fat-like differentiation, which induces energy to be burned in white fat as heat. This provides a new perspective for anti-obesity effect study of GN.

2) Why did the authors choose the 3T3-L1 cell line? Did the authors consider confirming their findings in primary adipocytes?

Answer: We appreciated for your advice. We used 3T3-L1 cells for the study of browning and obesity-related characteristics. This cell line is a well-established pre-adipose cell line that was developed from mouse embryos. 3T3-L1 presents an adipocyte-like phenotype under appropriate conditions. Moreover, it is easier to culture and less costly to use than isolated primary cells. Lastly, because 3T3-L1 can tolerate an increased number of passages and are homogeneous in terms of the cell population, so therefore we used 3T3-L1 [1,2].

3) The authors measured CO2 production in 3T3-L1 cells in response to treatment with GN and suggest that energy expenditure is increased in response to treatment. Did the authors consider using a Seahorse XF Analyser to confirm effects on the oxygen consumption rate? A robust response to these questions would be welcomed.

 Answer: We appreciated for your suggestion. We agree that energy expenditure have to include carbon dioxide emission and oxygen consumption. At the beginning of research, we wanted to measure oxygen consumption rate (OCR) in cells treated GN by using seahorse XF analyzer and fully considered this. However, because we didn’t have metabolic measurement analyzer, we could not analyze the metabolic change including oxygen consumption using it due to financial and technical problem. As a laboratory member for browning research, we have tried to prove the effect of dietary materials on energy expenditure and please understand that there are these limitations. Instead, we have suggested the mitochondrial activity by using mitotracker-red immunofluorescence analysis. Mitochondrial biogenesis activity is closely associated with energy metabolism in adipocytes. Regarding on it, we descripted as “These mitochondrial biogenesis activity is closely associated with energy metabolism by increasing increases oxygen consumption rate and β-oxidation in differentiated cells” in Discussion [3-5].

4) As a reader, I would like some indication of what GN is in the title.

 Answer: We appreciate for your advice. We changed the title as “Gonmisin N from Schisandra chinensis ameliorates lipid accumulation and induces a brown fat-like phenotype through AMP-Activated Protein Kinase in 3T3-L1 adipocytes”

5) Given that GN suppresses cell proliferation and cell cycle progression in the early stages of adipogenesis how do the authors account for this in their study? This needs to be further highlighted in the results/methods.

 Answer: We appreciate for your kind advice. We referred and designed a study in reference 28, which is determined the effect of GN on 3T3-L1 cells and HFD induced obese mice. In reference 28 [6], Gomisin N obtained from ChemFaces did not affect the cell viability in 3T3-L1 up to a high concentration of 10-100 µM, and reduced lipid accumulation by decreasing adipogenic gene expression. Moreover, they reported that GN ameliorates early stages of adipocyte differentiation rather than post stage. In our study, because GN showed cell toxicity at 50 µM which is a lower concentration than that of previous study, thereby cell treated with 6.25-25 µM of GN throughout differentiation duration. Therefore, we described this in Discussion.

6) Given the effects of treatment I would be interested to see the effect on FGF21 - was this measured? If so/possible, please report.

 Answer: We appreciated for your suggestion. As reviewer mentioned, fibroblast growth factor 21 (FGF21) plays a role as metabolic regulator in adipocytes. Thereby, we investigated the effect of Gomisin N on FGF21 expression in 3T3-L1 by Western blot analysis. As a result, GN stimulated FGF21 expression level. Therefore, we added this result in Figure 3, and addressed in Discussion.; “FGF21 also plays a role in energy metabolism [7]. FGF21 is preliminary expressed in adipose tissue, skeletal muscle or liver. Recent studies reported FGF21 increases energy expenditure by stimulating PGC1α and UCP1 expression, which are important for mitochondrial biogenesis in adipocytes [8].

7) Did the authors investigate how GN activates AMPK?

 Answer: Given the functional attributes of AMPK in lipid homeostasis, body weight, food intake, insulin signaling and mitochondrial biogenesis, AMPK is considered to be a major therapeutic target for the treatment of metabolic diseases including obesity [9,10]. In this study, 3T3-L1 cells were differentiated to BAT-like adipocytes and the differentiation medium of the white adipocytes was supplemented with 10μM AICAR, a known activator of AMPK. The activation of AMPK with AICAR led to the appearance of beige-like morphological properties including higher UCP1 expression differentiated white adipocytes as shown Figure4. In contrast, CC which is a known inhibitor of AMPK reduced UCP1 gene expression; however GN recovered UCP1 expression as much as control cells. Likewise, numerous naturally phytochemicals have been known to activate AMPK, such as resveratrol [11], quercetin [12], epigallocatechin gallate [13] and curcumim [14]. Mechanisms of activation of AMPK by these compounds appear to require the elevation of AMP levels because many of these compounds are known to inhibit mitochondrial ATP production. The mechanisms for AMPK activation by GN are largely unknown; however, presumably these compounds are likely to activate AMPK via AMP-dependent mechanisms.

8) The authors refer to an XTT assay in line 96 – had this been previously defined?

 Answer: We presented information of cell proliferation assay kit we used, in Method 4.2; “Cell viability was measured using a cell proliferation assay kit according to the manufacturer’s protocol (WelCount TR005-01, WelGENE, Seoul, Korea).”

9) Could the authors add a table of antibodies used in Western blotting studies (containing details of concentration and supplier)?

 Answer: We added information of antibodies in 4.1 Materials; “Antibodies against C/EBPα (sc-61), FAS (sc-20140), SREBP1c (sc-366), LPAATθ (sc-514164), DGAT1 (sc-32861), Lipin1 (sc-98450), PPARα (sc-9000), PPARγ (sc-7196), PGC1α (sc-13067), CPT1 (sc-393070), UCP1 (sc-6529) and GAPDH (sc-365062) were purchased from Santa Cruz Biotechnology, Inc. (Santa Cruz, CA, USA). Antibodies against p-ACC (cs-3661, Ser79) p-AMPK (cs-2535, Thr172), AMPK (cs-2603), p-HSL(cs-4139) and ACC (cs-3662) were purchased from Cell Signaling Technologies (Danvers, MA, USA). Antibodies against PPARδ (ab23673), PRDM16 (ab202344), UCP1 (ab23841), FGF21 (ab64857) and CPT1 (ab1285568) were purchased from Abcam (Cambridge, UK).”

10) The UCP1 (and PGC1α) blots in latter figures are highly variable – please can the authors account for this? For example, in figure 3 the UCP1 blot is ‘dirty’, whilst is relatively ‘clean’ in figure 4. Yet in figure 5, multiple bands are detected?

 Answer: We appreciate for pointing out Western blot analysis data. We suggest our original images of PGC1α and UCP1 data. Like below, PGC1α has been detected aging using same protein samples by polyclonal antibody (sc-13067), and its sensitivity was improved. In case of UCP1 analysis, we used two different UCP1 antibody; sc-6529 and ab23841. While UCP1 in Figure 3 was detected using sc-6529 which has lower sensitivity, UPC1 in Figure4 were obtained using ab23841. For better analyzing, we have purchased ab23841 at the end of the experiment. Also, we used the optimal concentration of polyacrylamide gel considering biomarker’s molecular weight. Lastly, we submitted all original images of Western data to editor by email, according to IJMS guideline.

Reference

  1. Ruiz-Ojeda, F.J.; Rupérez, A.I.; Gomez-Llorente, C.; Gil, A.; Aguilera, C.M. Cell models and their application for studying adipogenic differentiation in relation to obesity: a review. International journal of molecular sciences 2016, 17, 1040.
  2. Green, H.; Meuth, M. An established pre-adipose cell line and its differentiation in culture. Cell 1974, 3, 127-133.
  3. Si, Y.; Palani, S.; Jayaraman, A.; Lee, K. Effects of forced uncoupling protein 1 expression in 3T3-L1 cells on mitochondrial function and lipid metabolism. Journal of lipid research 2007, 48, 826-836.
  4. Lee, J.H.; Park, A.; Oh, K.-J.; Lee, S.C.; Kim, W.K.; Bae, K.-H. The Role of Adipose Tissue Mitochondria: Regulation of Mitochondrial Function for the Treatment of Metabolic Diseases. International journal of molecular sciences 2019, 20, 4924.
  5. Kusminski, C.M.; Scherer, P.E. Mitochondrial dysfunction in white adipose tissue. Trends Endocrinol Metab 2012, 23, 435-443, doi:10.1016/j.tem.2012.06.004.
  6. Park, H.J.; Cho, J.-Y.; Kim, M.K.; Koh, P.-O.; Cho, K.-W.; Kim, C.H.; Lee, K.-S.; Chung, B.Y.; Kim, G.-S.; Cho, J.-H. Anti-obesity effect of Schisandra chinensis in 3T3-L1 cells and high fat diet-induced obese rats. Food Chemistry 2012, 134, 227-234.
  7. Badman, M.K.; Pissios, P.; Kennedy, A.R.; Koukos, G.; Flier, J.S.; Maratos-Flier, E. Hepatic fibroblast growth factor 21 is regulated by PPARα and is a key mediator of hepatic lipid metabolism in ketotic states. Cell metabolism 2007, 5, 426-437.
  8. Chau, M.D.; Gao, J.; Yang, Q.; Wu, Z.; Gromada, J. Fibroblast growth factor 21 regulates energy metabolism by activating the AMPK–SIRT1–PGC-1α pathway. Proceedings of the National Academy of Sciences 2010, 107, 12553-12558.
  9. Kim, J.; Yang, G.; Kim, Y.; Kim, J.; Ha, J. AMPK activators: mechanisms of action and physiological activities. Experimental & molecular medicine 2016, 48, e224-e224.
  10. Abdul-Rahman, O.; Kristóf, E.; Doan-Xuan, Q.-M.; Vida, A.; Nagy, L.; Horváth, A.; Simon, J.; Maros, T.; Szentkirályi, I.; Palotás, L. AMP-activated kinase (AMPK) activation by AICAR in human white adipocytes derived from pericardial white adipose tissue stem cells induces a partial beige-like phenotype. PLoS One 2016, 11, e0157644.
  11. Baur, J.A.; Pearson, K.J.; Price, N.L.; Jamieson, H.A.; Lerin, C.; Kalra, A.; Prabhu, V.V.; Allard, J.S.; Lopez-Lluch, G.; Lewis, K. Resveratrol improves health and survival of mice on a high-calorie diet. Nature 2006, 444, 337-342.
  12. Ahn, J.; Lee, H.; Kim, S.; Park, J.; Ha, T. The anti-obesity effect of quercetin is mediated by the AMPK and MAPK signaling pathways. Biochemical and biophysical research communications 2008, 373, 545-549.
  13. Hwang, J.-T.; Park, I.-J.; Shin, J.-I.; Lee, Y.K.; Lee, S.K.; Baik, H.W.; Ha, J.; Park, O.J. Genistein, EGCG, and capsaicin inhibit adipocyte differentiation process via activating AMP-activated protein kinase. Biochemical and biophysical research communications 2005, 338, 694-699.
  14. Kim, T.; Davis, J.; Zhang, A.J.; He, X.; Mathews, S.T. Curcumin activates AMPK and suppresses gluconeogenic gene expression in hepatoma cells. Biochemical and biophysical research communications 2009, 388, 377-382.

Reviewer 3 Report

(1) Authors here studied the effects of gomisin N (GN) isolated from Schisandra chinensis, a traditional herbal sedative and tonic agent, on lipid accumulation and adipocyte browning during adipocyte differentiation. In this manuscript, they report that GN reduces the triglyceride content of 3T3-L1 cells and induces browning of 3T3-L1 adipocytes by decreasing expression of adipogenesis genes and by increasing expression of fatty acid oxidation-related and thermogenic genes, suggesting that GN have therapeutic potential in the treatment of obesity and obesity-associated metabolic disease. It is also reported that GN induces its effect through activation of AMP-activated protein kinase (AMPK).

(2) The conclusions are well supported by the reported results and interesting; however, the manuscript contains several possible errors. The following should be confirmed by the authors (suggestions or comments are indicated by "-->"):

p. 3, lines 54- 55:
fibroblast-associated (FAS)
-->? fatty acid synthase (FAS)

p. 5, line 110:
by up-regulating lipogenic factors
-->? by down-regulating lipogenic factors

p. 13, line 309; p. 19, line 497; p. 23, line 531:
18s rRNA
--> 18S rRNA
(Comment: S, Svedberg)

p. 17, lines 458 and 480:
--> Comment: Reference information is incomplete.

p. 19, line 501:
--> Comment: In Figure 1F, "GN (μM)" should be added to the title of the horizontal axis.

p. 21, line 510:
--> Comment: The titles of Figures 2 and 3 are the same. The title of Figure 3 may be changed to "Effects of GN on expression of thermogenic markers and fatty acid oxidation-related genes in 3T3-L1 cells".

p. 21, lines 510-515:
--> Comment: There is no explanation for panel D.

(3) During the review, I also noticed several minor points to be confirmed by the authors:

p. 3, line 37:
divided two major types:
--> divided into two major types:

p. 4, line 59:
AMPK
--> AMP-activated protein kinase (AMPK)

p. 4, lines 78-79:
However, the effects of GN on thermogenic activity and FAO and subsequent inhibition of lipid accumulation in adipocytes has not been explored.
--> However, the effects of GN on thermogenic activity and FAO and subsequent inhibition of lipid accumulation in adipocytes have not been explored.

p. 5, line 93:
respectively
--> respectively

p. 5, line 106:
sterol regulatory element-binding transcription factor 1 (SREBP1c),
-->? sterol regulatory element-binding transcription factor 1c (SREBP1c),

p. 6, line 135:
results to inactivation of ACC
--> results in inactivation of ACC

p. 7, line 143:
BMP4
--> BMP4 (20 ng/mL)

p. 10, line 226:
gonmisin
--> gonmisin

p. 10, lines 249-250:
, recombinant human BMP4 and trypsin ethylenediaminetetraacetic acid (EDTA) were
-->? , recombinant human BMP4, and trypsin and ethylenediaminetetraacetic acid (EDTA) were

p. 11, line 256:
Antibody
--> Antibodies

p. 11, lines 261-264:
--> Comment: The reference(s) for the XTT assay must be cited or assay conditions, such as XTT and PMS concentrations, must be provided.

p. 11, line 271:
After a further 2 days,
--> After a further culture for 2 days

p. 11, line 274:
by 2 days treatment
--> by two days treatment

p. 11, line 276:
5 mg/mL insulin
-->? 5 μg/mL insulin

p. 11, line 276:
Compound GN stock was prepared in dimethyl sulfoxide
--> A xx mM GN stock solution was prepared in dimethyl sulfoxide

p. 12, line 291:
protease inhibitor cocktail
--> Comment: The "protease inhibitor cocktail" should be specified.

p. 13, line 315:
protease inhibitors
--> Comment: The "protease inhibitors" should be specified.

p. 13, line 319:
with TBS-T buffer.
--> with Tris-buffered saline with Tween 20 (TBS-T) buffer

p. 13, line 320:
were immunoblotted with primary antibodies (1:1,000) overnight at 4°C.
-->? were immunoblotted with primary antibodies (1:1,000) overnight at 4°C and washed with TBS-T buffer.

p. 13, lines 321-322:
were incubated with secondary antibodies conjugated to horseradish peroxidase (1:2,000) at room temperature for 6 h.
-->? were incubated with secondary antibodies conjugated to horseradish peroxidase (1:2,000) at room temperature for 6 h and washed with TBS-T buffer.

p. 18, line 490:
18s
-->? 18S

p. 19, line 500:
--> Comment: The abbreviations CD and ND in Figure 1 must be explained.

p. 22, line 518:
3T3-L1 were treated
--> 3T3-L1 cells were treated

p. 22, line 519:
10μM AICAR ± GN,
--> 10μM AICAR ± 25μM GN,

p. 22, line 520:
10μM CC ± GN,
--> 10μM CC ± 25μM GN,

p. 24, line 538:
20 ng/ml BMP4 ± 25μM GC,
--> 20 ng/mL BMP4 ± 25μM GN,

Author Response

Dear reviewer,

Thank you for considering our manuscript for publication in International Journal of Molecular Sciences. We are very pleasure to have been given the opportunity to revise our manuscript, “Gomisin N from Schisandra chinensis ameliorates lipid accumulation and induces a brown fat-like phenotype through AMP-Activated Protein Kinase in 3T3-L1 adipocytes”. We have addressed the reviewer’s comments point-by-point and made the necessary changes to the manuscript. Please see the attachment.

We hope that the manuscript is acceptable for publication in International Journal of Molecular Sciences. We appreciate again to consider this paper for publication in IJMS, and declare that authors of this work have no conflict of interests.

Sincerely, yours,

Boo-Yong Lee, Ph.D.

Reviewer 3

1) p. 3, lines 54- 55: fibroblast-associated (FAS)-->? fatty acid synthase (FAS)

Answer: Thank you for your pointing out our mistake. We revised as “fatty acid synthase (FAS)”.

2) p. 5, line 110:by up-regulating lipogenic factors-->? by down-regulating lipogenic factors

Answer: As reviewer’s comment, we revised as “down-regulating”. Thank you for your pointing out our mistake.

3) p. 13, line 309; p. 19, line 497; p. 23, line 531: 18s rRNA--> 18S rRNA (Comment: S, Svedberg)

Answer: As reviewer’s comment, we revised as “18S rRNA”.

4) p. 17, lines 458 and 480:--> Comment: Reference information is incomplete.

Answer: We revised the reference information [39-46].

5) p. 19, line 501: --> Comment: In Figure 1F, "GN (μM)" should be added to the title of the horizontal axis.

Answer: We added the title of the horizontal axis as GN (μM) in Figure 1F.

6) p. 21, line 510: --> Comment: The titles of Figures 2 and 3 are the same. The title of Figure 3 may be changed to "Effects of GN on expression of thermogenic markers and fatty acid oxidation-related genes in 3T3-L1 cells".

Answer: As reviewer’s comment, we changed the title as “Effects of GN on expression of thermogenic markers and fatty acid oxidation-related genes in 3T3-L1 cells”. Thank you for your pointing out our mistake.

7) p. 21, lines 510-515: --> Comment: There is no explanation for panel D.

Answer: We added “(D) Immunofluorescence image of 3T3-L1 (400× magnification)”.

8) p. 3, line 37: divided two major types:--> divided into two major types:

Answer: We revised as “divided into two major types:”.

9) p. 4, line 59: AMPK--> AMP-activated protein kinase (AMPK)

Answer: We added full name of AMPK on page4, line 60.

10) p. 4, lines 78-79: However, the effects of GN on thermogenic activity and FAO and subsequent inhibition of lipid accumulation in adipocytes has not been explored. --> However, the effects of GN on thermogenic activity and FAO and subsequent inhibition of lipid accumulation in adipocytes have not been explored.

Answer: We revised as “However, the effects of GN on thermogenic activity and FAO and subsequent inhibition of lipid accumulation in adipocytes have not been explored.”.

11) p. 5, line 93:respectively--> respectively

Answer: We removed “respectively” in this sentence.

12) p. 5, line 106: sterol regulatory element-binding transcription factor 1 (SREBP1c),-->? sterol regulatory element-binding transcription factor 1c (SREBP1c),

Answer: According to your advice, we revised the full name of SREBP1c exactly.

13) p. 6, line 135:results to inactivation of ACC --> results in inactivation of ACC

Answer: We revised as “results in inactivation of ACC”.

14) p. 7, line 143:BMP4--> BMP4 (20 ng/mL)

Answer: We presented the exact concentration of BMP4.

15) p. 10, line 226: gonmisin--> gonmisin

Answer: We re-wrote “gomisin” in this sentence.

16) p. 10, lines 249-250: , recombinant human BMP4 and trypsin ethylenediaminetetraacetic acid (EDTA) were-->? , recombinant human BMP4, and trypsin and ethylenediaminetetraacetic acid (EDTA) were

Answer: We revised as “, recombinant human BMP4, trypsin and ethylenediaminetetraacetic acid (EDTA) were”.

17) p. 11, line 256:Antibody--> Antibodies

Answer : We re-wrote “Antibodies” in this sentence.

18) p. 11, lines 261-264: --> Comment: The reference(s) for the XTT assay must be cited or assay conditions, such as XTT and PMS concentrations, must be provided.

Answer: We presented information of cell proliferation assay kit in Method 4.2; “Cell viability was measured using a cell proliferation assay kit according to the manufacturer’s protocol (WelCount TR005-01, WelGENE, Seoul, Korea).”

19) p. 11, line 271:After a further 2 days,--> After a further culture for 2 days

Answer: According to your comment, we revised as “After a further culture for 2 days”.

20) p. 11, line 274:by 2 days treatment--> by two days treatment

Answer: : We re-wrote “by two days treatment” in this sentence.

21) p. 11, line 276:5 mg/mL insulin-->? 5 μg/mL insulin

Answer: We revised the exact unit as 5 μg/mL insulin.

22) p. 11, line 276: Compound GN stock was prepared in dimethyl sulfoxide --> A xx mM GN stock solution was prepared in dimethyl sulfoxide

Answer: We revised as “A 50mM GN stock solution was prepared in dimethyl sulfoxide ”.

23) p. 12, line 291:protease inhibitor cocktail--> Comment: The "protease inhibitor cocktail" should be specified.

Answer: We addressed the information of protease inhibitor cocktail in Method 4.5; “Cells were lysed using lysis buffer (iNtRON Biotechnology, Seoul, Korea) with protease inhibitor cocktail (Biomake B14001, Houston, USA), phosphatase inhibitor 2 (Sigma P5726) and phosphatase inhibitor 3 (Sigma P0044).”

 24) p. 13, line 315:protease inhibitors--> Comment: The "protease inhibitors" should be specified.

Answer: We addressed the information of phosphatase inhibitor in Method 4.5; Cells were lysed using lysis buffer (iNtRON Biotechnology, Seoul, Korea) with protease inhibitor cocktail (Biomake B14001, Houston, USA), phosphatase inhibitor 2 (Sigma P5726) and phosphatase inhibitor 3 (Sigma P0044).”

25) p. 13, line 319:with TBS-T buffer.--> with Tris-buffered saline with Tween 20 (TBS-T) buffer

Answer: According to your kind advice, we added the full name of TBS-T in this sentence.

26) p. 13, line 320:were immunoblotted with primary antibodies (1:1,000) overnight at 4°C.
-->? were immunoblotted with primary antibodies (1:1,000) overnight at 4°C and washed with TBS-T buffer.

Answer: We revised this sentence as “Membranes were immunoblotted with primary antibodies (1:1,000) overnight at 4°C and washed with TBS-T buffer”

27) p. 13, lines 321-322:were incubated with secondary antibodies conjugated tohorseradish peroxidase (1:2,000) at room temperature for 6 h.-->? were incubated with secondary antibodies conjugated to horseradish peroxidase (1:2,000) at room temperature for 6 h and washed with TBS-T buffer.

Answer: We revised this sentence as “Subsequently, membranes were incubated with secondary antibodies conjugated to horseradish peroxidase (1:2,000) at room temperature for 6 h and washed with TBS-T buffer”

28) p. 18, line 490:18s-->? 18S

Answer: We changed all of 18s into 18S.

 29) p. 19, line 500:--> Comment: The abbreviations CD and ND in Figure 1 must be explained.

Answer: We presented information of ND and CD in Figure 1: ND; non-differentiation. CD; control-differentiation.

30) p. 22, line 518:3T3-L1 were treated--> 3T3-L1 cells were treated

Answer: We revised as “3T3-L1 cells were treated” in Figure 4.

31) p. 22, line 519:10μM AICAR ± GN,--> 10μM AICAR ± 25μM GN,

Answer: We addressed the exact concentration of GN in Figure 4.

32) p. 22, line 520:10μM CC ± GN,--> 10μM CC ± 25μM GN,

Answer: Likewise, we also addressed the exact concentration of GN in Figure 4.

33) p. 24, line 538: 20 ng/ml BMP4 ± 25μM GC, --> 20 ng/mL BMP4 ± 25μM GN,

Answer: We revised as “20 ng/mL BMP4 ± 25μM GN” in Figure 6.
